# MSdeCIpher: A Tool to Link Data from Complementary Ionization Techniques in High-Resolution GC-MS to Identify Molecular Ions

**DOI:** 10.3390/metabo14010010

**Published:** 2023-12-22

**Authors:** Daniel Stettin, Georg Pohnert

**Affiliations:** 1Institute for Inorganic and Analytical Chemistry, Bioorganic Analytics, Friedrich Schiller University Jena, 07743 Jena, Germany; daniel.stettin@uni-jena.de; 2Cluster of Excellence Balance of the Microverse, Friedrich Schiller University Jena, 07743 Jena, Germany

**Keywords:** high-resolution mass spectrometry (HRMS), gas chromatography (GC), ionization technique, molecular ion, software tool, analyte identification

## Abstract

Electron ionization (EI) and molecular ion-generating techniques like chemical ionization (CI) are complementary ionization methods in gas chromatography (GC)-mass spectrometry (MS). However, manual curation effort and expert knowledge are required to correctly assign molecular ions to fragment spectra. MSdeCIpher is a software tool that enables the combination of two separate datasets from fragment-rich spectra, like EI-spectra, and soft ionization spectra containing molecular ion candidates. Using high-resolution GC-MS data, it identifies and assigns molecular ions based on retention time matching, user-defined adduct/neutral loss criteria, and sum formula matching. To our knowledge, no other freely available or vendor tool is currently capable of combining fragment-rich and soft ionization datasets in this manner. The tool’s performance was evaluated on three test datasets. When molecular ions are present, MSdeCIpher consistently ranks the correct molecular ion for each fragment spectrum in one of the top positions, with average ranks of 1.5, 1, and 1.2 in the three datasets, respectively. MSdeCIpher effectively reduces candidate molecular ions for each fragment spectrum and thus enables the usage of compound identification tools that require molecular masses as input. It paves the way towards rapid annotations in untargeted analysis with high-resolution GC-MS.

## 1. Introduction

Mass spectrometry has become a cornerstone of biochemical analytics [1]. In recent years, the prospect of compound identification solely via mass spectrometry has increased in importance but remains a challenging task in a wide array of research fields, ranging from plant sciences [2] to foods [3], pesticide residue [4], disease biomarkers [5], and drug discovery [6]. The discovery of novel compounds, as well as the dereplication of known compounds, in complex biological samples hinges on the ability to correctly predict chemical structures via mass spectrometry [7].

Mass spectrometry hyphenated to gas chromatography has a wide range of applications in research and industry. Using the standard 70 eV electron ionization (EI), it provides reproducible spectra with a vast amount of spectral library support [8]. Even when no direct spectral match is available, computational tools exist that can evaluate the likelihood of a putative structure (MS-Finder [9], CFM-ID [10], MetExpert [11]) by utilizing the accurate mass capabilities of modern instrumentation. The first step of the workflow of those identification tools is to query the accurate molecular mass of the unknown compound against compound databases [12]. However, EI spectra are often devoid of a molecular ion [13], so the molecular mass of an unknown compound is often impossible to obtain this way. Even high-resolution GC-EI-MS is, on its own, insufficient when trying to identify unknowns in cases where no fragment library match is available [14]. Alternative soft ionization techniques, which limit the amount of fragmentation by exposing the analytes to less excess energy during ionization, such as chemical ionization (CI) [15] or atmospheric pressure chemical ionization (APCI) [16], are required to generate molecular ions instead.

To get the best of both worlds in untargeted screening efforts, EI and a molecular ion-generating ionization technique have to be used in parallel. This has been applied successfully in a number of studies to identify previous unknowns from extracts of *Escherichia coli*, *Chlamydomonas*, and *Artemisia* [17], a diverse set of human, animal, and marine samples [18], *Saccharomyces cerevisiae* [19], and *Skeletonema costatum* [14]. It has furthermore been used to increase metabolomic coverage in human serum samples [20], to identify forensically relevant compounds [21,22], to perform non-targeted analysis of environmental pollutants [23], and to identify compounds from cometary ice [24].

The problem that researchers face when using this approach is that the two complementary ionization techniques result in two separate datasets from two independent runs. Information from these two datasets is not easy to combine. Retention time shifts between the two runs are possible and ionization rates per compound will often change between the two techniques, yielding chromatograms that differ in appearance. The development of gas chromatography hyphenated to EI and CI simultaneously is underway [25], potentially simplifying retention time alignment between the two datasets. Still, mass spectra differ in amount and mechanism of fragmentation, resulting in substantial differences per compound [26]. In consequence, the assignment of molecular ions in one dataset to the fragment-rich EI spectra in the other dataset has to be done manually with expert knowledge for every single compound.

Exploiting the mass accuracy and high-resolution capabilities that modern mass analyzers like the Orbitrap offer, it is possible to develop a computational strategy that automates this process. Given constraints on elements possibly contained in an analyte, high-resolution data can be used to calculate sum formulas for ions with reasonable accuracy, leaving few possible candidate sum formulas per ion. Exploiting the fact that all fragments of a molecular ion must contain a subset of that ion’s sum formula (disregarding edge cases of adduct formation during fragmentation), the most likely sum formula of a molecular ion can be computationally elucidated by considering all possible sum formulas of all fragments. Additionally, because molecular ions and fragments are in two different datasets and not linked, this approach can also be used to establish a candidate’s identity as a molecular ion for a specific fragment spectrum.

We developed the easy-to-use software tool MSdeCIpher in the coding language R with a graphical user interface that enables the automated identification and assignment of molecular ions to their respective fragment-rich spectra. MSdeCIper contains the embedded abbreviations “MS” (Mass Spectrometry) and “CI” (Chemical Ionization), as well as “decipher”, as a play on the tool’s ability to “decipher” the difficult-to-elucidate connection between fragment-rich and soft ionization spectra from two ionization techniques. It has been developed for users of GC-HRMS pipelines that rely on electron ionization for strong library support but want to increase their capability of identifying unknowns by integrating a molecular ion-generating technique into their workflow.

We performed the evaluation of this tool using high-resolution GC-Orbitrap CI and EI spectra, but the tool is also compatible with other techniques, such as APCI ionization without adjustment.

## 2. Materials and Methods

### 2.1. Analytical Standards

Metabolite standards in Table 1 from the top until entry succinate were taken from the Mass Spectrometry Metabolite Library of Standards by IROA technologies (Sigma-Aldrich, Munich, Germany). 5-Oxo-L-proline, α-tocopherol, cholesta-3,5-diene, cholesterol, glycero-1-phosphate, glycero-2-phosphate, urea, L-methionine, L-rhamnose, myo-inositol, phosphoric acid, phytol, and xylitol were obtained from Sigma-Aldrich, Munich, Germany. D-Mannose, L-arabitol, and spermine were obtained from Alfa Aesar, Kandel, Germany. D-Allose was obtained from Carl Roth, Karlsruhe, Germany. Docosahexaenoic acid was obtained from Acros Organics, Geel, Belgium. Glycine, L-lysine, L-serine, and L-tyrosine were obtained from Fluka Analytical, Seelze, Germany. Scyllo-inositol was obtained from abcam Biochemicals, Berlin, Germany. Organophosphorous pesticide standards were obtained as EPA 8270 Organophosphorus Pesticide Mix 2 (Sigma-Aldrich, Munich, Germany).

### 2.2. Sample Preparation

Metabolite standards (5 µg each) were taken up in 100 µL methanol each (LiChrosolv, Merck, Darmstadt, Germany), dried under vacuum overnight, and reconstituted in 20 µL pyridine (Sigma-Aldrich, Munich, Germany) containing 20 mg/mL methoxyamine monohydrochloride (Sigma-Aldrich, Munich, Germany). After heating at 60 °C for 1 h and storage at room temperature overnight, 20 µL of N,O-bis(trimethylsilyl)trifluoroacetamide (BSTFA) (Thermo Scientific, Bremen, Germany) was added to each sample, and all samples were heated to 60 °C for 1 h again.

Organophosphorus pesticide standards were diluted 1:10 in dichloromethane (Sigma-Aldrich, Munich, Germany) to a final concentration of 200 µg/mL per component.

### 2.3. Data Acquisition

Metabolite standard and organophosphorus pesticide standard datasets were acquired on a Q-Exactive™ Orbitrap™ GC system, consisting of a Q Exactive™ Orbitrap™ mass spectrometer and a Trace™ 1310 GC equipped with a TriPlus™ RSH™ Autosampler (Thermo Scientific, Bremen, Germany). The GC was equipped with Zebron ZB-SemiVolatiles columns (30 m × 0.25 mm × 0.25 µm, Phenomenex, Aschaffenburg, Germany). The injection temperature was kept at 250 °C. The injection volume for all samples was 1 µL. The carrier gas flow was kept at 1 mL/min. For metabolite standards, a split ratio of 1:25 was used in EI and splitless mode with a splitless time of 2 min was used in CI. For organophosphorus standards, a split ratio of 1:100 was used in EI and 1:20 was used in CI. Metabolite standards were measured with an oven temperature program starting at 80 °C, maintained for 2 min, raised to 320 °C at a rate of 100 °C/min, and maintained for 2 min. The organophosphorus pesticide mix was measured starting at 80 °C, maintained for 2 min, and raised to 320 °C at a rate of 20 °C/min. The transfer line was kept at 250 °C. The ion source was kept at 300 °C in EI mode and at 180 °C in CI mode. Methane (N55, Air Liquide, Düsseldorf, Germany) at a flow rate of 1.5 mL/min was used as the ionization gas in CI. Data were recorded in full scan profile mode at a Fourier transform resolution of 120,000. Scan range was set to 50–600 *m*/*z* in EI and 80–1200 *m*/*z* in CI.

The metabolomics dataset was obtained from a previous study [14].

### 2.4. Data Deconvolution

Data were preprocessed as described in a previous study [14]. Acquired raw data files were converted to mzXML format using MSConvert (proteowizard.sourceforge.net (accessed on 21 December 2023)). Deconvolution was achieved via a custom R pipeline based on the packages XCMS [27], CAMERA [28], and metaMS [29]. In short, XCMS performed initial feature deconvolution, and CAMERA performed grouping of extracted features into extracted spectra (pseudospectra). A custom script then filtered out pseudospectra with too few fragments for analysis. metaMS was then used to create a file compatible with the NIST library from all pseudospectra. The exact script can be found in the Appendix A of the previous study [14]. When single files needed to be deconvoluted, a modified version of this R script was used (available in the Appendix A). A summary of XCMS and CAMERA parameters is available in Appendix A.

It is important to note that MSdeCIpher does not require this particular method of data deconvolution to be used. Any deconvolution pipeline can be used, as long as deconvoluted data from the EI and soft ionization runs including *m*/*z*, retention time, intensity, and pseudospectra assignment of each individual feature can be provided. Check https://github.com/Pohnert-Lab/MSdeCIpher (accessed on 21 December 2023) for example files and a tutorial on the required data format.

### 2.5. MSdeCIpher Settings

The following settings were used for every MSdeCIpher analysis: Mass accuracy 3 ppm; minimum number of *m*/*z* values 20 for both EI and CI; how many *m*/*z* differences need to be found—2; additional filtering based on *m*/*z*; top x candidates—10; retention time tolerance 0.05 min; raw data for adduct/fragment search and sum formula correction enabled.

The following settings differed depending on the dataset: *m*/*z* differences −16.03130, 28.03130, and 40.03130 for metabolomics and metabolite standard datasets, and 28.03130 and 40.03130 for organophosphorus standard dataset; element constraints C 0 to 50, H 0 to 50, N 0 to 50, O 0 to 50, S 0 to 50, Si 0 to 50, and P 0 to 50 for metabolomics and metabolite standard datasets, and C 0 to 50, H 0 to 50, N 0 to 50, O 0 to 50, S 0 to 50, Cl 0 to 50, and P 0 to 50 for organophosphorus standard datasets; use retention time standards enabled for the metabolomics dataset, disabled for standard datasets.

MSdeCIpher’s source code and a tutorial for parameter usage can be obtained from https://github.com/Pohnert-Lab/MSdeCIpher (accessed on 21 December 2023).

## 3. Results

MSdeCIpher is a software tool designed to assign possible molecular ions to fragment spectra in two separate GC-HRMS datasets acquired with different ionization techniques, one dataset containing fragment spectra (EI) and the other containing possible molecular ions (i.e., CI). It was written in the programming language R and comes with an easy-to-use shiny interface (Figure 1). It takes 3.5 h to process the example dataset (483 compounds over a 40 min runtime) on a PC with an AMD Ryzen 7 3800X (8^x3^.9 GHz) and 16 GB RAM.

### 3.1. Workflow

MSdeCIpher uses deconvoluted peak lists of both datasets as a starting point. These peak lists need to contain all deconvoluted features with their individual accurate *m*/*z*, retention time, integrated area, and assigned chromatographic peak group (pseudospectrum). All freely available and vendor deconvolution tools can be used as long as they can produce this output.

The first step in MSdeCIpher’s workflow (Figure 1) is filtering unusable data from these peak lists. Depending on user input parameters, all pseudospectra that do not contain a defined minimum number of features are deleted.

Both datasets are connected via retention time matching. Each fragment pseudospectrum gets assigned none, one, or multiple pseudospectra from the molecular ion dataset. The size of the retention time window in which this matching takes place is dependent on user input. Optionally, a table containing retention times of retention time standards that appear in both datasets can be used to correct for retention time shift between the two runs.

Each pseudospectrum from the molecular ion dataset (here CI spectra) is then searched for potential molecular ion candidates. Adduct and neutral loss criteria that molecular ions are expected to display can be defined as input parameters (Figure 2). Depending on the ionization technique, this can facilitate the identification of the correct [M + H]^+^ molecular ion species [30], and reduces the peak lists from the molecular ion dataset to fewer candidate ions per pseudospectrum. Because the intensity of expected adduct and neutral loss ions can sometimes be extremely low and thus not likely to be picked up by the deconvolution tool used, MSdeCIpher also offers the option to perform the search in the raw data file instead of the deconvoluted input data.

After this data treatment, many candidates remain (Figure 3 red arrows). This list can be further refined by deleting candidates with low *m*/*z*. This is allowed since signals with the highest *m*/*z* are the most likely candidates for molecular ions. In the case of APCI-MS, one can also select for ions with high intensity since molecular ions often dominate the spectra.

MSdeCIpher then calculates sum formulas of all fragment ions and molecular ion candidates with the RCDK package [32], an R implementation of the Chemistry Development Kit [33]. Prior to this, M + 1 and M + 2 isotopic peaks are removed from the pseudospectra for the purpose of sum formula calculation (isotopic pattern analysis is performed independently in raw data as described below).

Left with few candidate molecular ions from one or multiple pseudospectra, the assumption used by MSdeCIpher is that the sum formula of the true molecular ion should include the sum formulas of all fragment ions from the fragment pseudospectrum. Thus, the candidate with a sum formula that is supported by the most fragments is most likely to be the true molecular ion of the compound. However, a prerequisite for this workflow is the correct assignment of sum formulas to each ion. This is not easily achievable since even high-resolution instrumentation like the Orbitrap MS is not able to reduce candidate sum formulas to a single possibility for most ions [34].

To circumvent this, MSdeCIpher uses a “bottom-up” approach to statistically narrow down multiple possible sum formulas for an ion to the most likely correct one. Because the mass accuracy of an Orbitrap is relative to the *m*/*z* of an ion (parts per million), the absolute Δ*m*/*z* uncertainty is the smallest for low *m*/*z* ions. That means that small ions (<100 *m*/*z*) usually only have one possible sum formula, given a mass accuracy of ~2 Δppm and a constrained set of possible elements like CHNOPSSi. As soon as an ion yields multiple possible sum formulas, all possibilities are evaluated in light of the previously assigned sum formulas for smaller ions in the same pseudospectrum. Every retained sum formula is connected with a weighted score depending on its *m*/*z* and intensity. MSdeCIpher presents this score as a probability score in percent. It is a measure of the percentage of intensity in the fragment pseudospectrum that supports the sum formula, with the summed-up intensity of all fragments that were successfully assigned a sum formula in a fragment pseudospectrum being equal to 100%. However, fragment spectra intensity is often dominated by one or a few fragments. Also, higher *m*/*z* fragment ions are per se more informative than lower *m*/*z* ions for evaluating molecular ion candidates even though they might be of lower intensity. For that reason, the relative intensity (Int_rel_) contributing to the score by each fragment ion is log-scaled and weighted based on its *m*/*z* value according to:(1)$$m/z\times ln({Int}_{rel}\times 100+1)$$
inspired by a similar formula by Hufsky et al. [35]. This decreases the relative contribution of high intensity fragment ions and increases the relative contribution of high *m*/*z* fragment ions.

The highest scoring sum formula is then retained and will in turn be used again to evaluate the next set of possible sum formulas. This chain continues until all fragments have been assigned one sum formula. In the case that none of the possible sum formulas of an ion fit any of the previously assigned ones (i.e., noise peaks or a valid fragment with different elements than previous smaller fragments), all possible sum formulas are retained. Molecular ion candidates can then be assigned a sum formula in the same manner and at the same time be given a probability rating based on how much the fragment-score supports each possible molecular ion (Figure 4).

Additionally, to increase confidence in sum formula assignment, multiple measures are employed to refine the list of possible sum formulas for an ion, taken from the “Seven Golden Rules” [36]. The default element constraints input recommended by the MSdeCIpher interface is based around the maximum number of elements in sum formulas below 500 Da presented therein. Also, heuristic filtering is implemented in MSdeCIpher, restricting possible element ratios in sum formulas to common ranges. A simplified version of isotope pattern analysis is also part of MSdeCIpher, when raw data are provided by the user. It exploits the high-resolution capabilities of the Orbitrap MS to resolve the detailed pattern of isotopic peaks. A check is performed to assess whether the isotopic peaks expected for the elements in the proposed sum formula do indeed appear in the isotopic pattern of the peak in question. While this is not very useful for C, H, N, and O because of their ubiquitous appearance in compounds, rarer elements like S, Cl, or Br can be effectively ruled out when their distinct isotopic peaks are missing. MSdeCIpher performs those checks in raw data to make sure elements are not falsely ruled out because of underperforming deconvolution.

This results in a ranking of molecular ion candidates for each fragment pseudospectrum according to their probability score.

### 3.2. Performance Evaluation

MSdeCIpher’s performance was evaluated with an Orbitrap GC-MS on two datasets comprising analytical standards and one “real-world” dataset.

Since compound identification is a key topic in metabolomics research [7], metabolite standards were chosen as the first evaluation dataset. Standards were only processed further if they displayed a visible TIC peak above the baseline in both EI and CI modes. Most molecular ions were assigned correctly by MSdeCIpher and appeared within the first few places of the ranking, with an average rank of 1.5 of the correct molecular ion across all standards where a molecular ion was present in the deconvoluted data (Table 1). In 9 out of the 32 compounds, the assignment was not possible due to a missing molecular ion or missing adduct/neutral loss pattern.

Also, in the field of residue analysis, the generation of molecular ions with accurate mass instrumentation becomes increasingly more important when using untargeted approaches [4,37]. Therefore, organophosphorus pesticides were chosen as the second evaluation dataset. They are reported to typically have a low abundance of molecular ions in EI which hampers easy identification and quantification [38]. Here, all molecular ions were assigned correctly in rank 1 (Table 2).

The third dataset contained measured biological extracts from a previous study, a metabolomics experiment with the microalgae *Skeletonema costatum* [14]. It was chosen to assess the tool’s performance in real-world samples in a peak-rich environment (45714 features comprising 483 compounds over a 40 min runtime). To create a benchmarking dataset for MSdeCIpher, we attempted to identify all 483 compounds with library matching as described in the original publication [14]. That way, 40 out of the 483 chromatographic peaks could be identified at the MSI 1 level (confirmation with an analytical standard), with 37 of those used for benchmarking. In the remaining three cases, identification with a standard was successful, but the exact molecular species, i.e., derivatization state, and thus molecular mass, could not be determined. MSdeCIpher assigned the correct molecular ion in 24 of the 37 benchmark compounds with an average rank of 1.2 (Table 3). In the remaining 13 cases, the assignment was not possible due to a missing molecular ion, missing adduct/neutral loss pattern, or an overall intensity of the CI spectrum that is too low.

## 4. Discussion

Our tool yields good to excellent assignments of the molecular ions in all test datasets (Table 1, Table 2 and Table 3). However, when using MSdeCIpher, it is important to keep certain limitations in mind. For one, MSdeCIpher will always be limited by the efficacy of the ionization method used. In certain cases, no molecular ion is observed or the gas phase chemistry does not give adducts that are defined in our tool in the “adduct/neutral loss” section. MSdeCIpher has no way of predicting such a behavior and will present a candidate list devoid of the true molecular ions, as it is observed in our datasets (see commented entries in Table 1 and Table 3). These are problems that cannot be overcome with algorithmic solutions but are rather unavoidable imperfections of the underlying chromatography and mass spectrometry.

Secondly, in cases where the true molecular ion was not listed first, but in the top four, co-eluting compounds of substantially higher molecular weight were observed. Because of the way MSdeCIpher evaluates the plausibility of sum formulas, such molecular ion candidates with high *m*/*z* will receive higher scores by default than smaller but true molecular ions. In the case of such a co-elution, manual curation of the data is required.

It is, therefore, recommended that the user treats the few top results with comparable scores as a putative candidate list to be used either in a subsequent identification pipeline or pending manual review. MSdeCIpher is meant to be used for hypothesis generation, not validation.

## 5. Conclusions

MSdeCIpher successfully combines fragment- and molecular ion-containing datasets obtained with high-resolution GC-MS systems, providing a candidate list of molecular ions for each chromatographic peak. When molecular ions are present, MSdeCIpher consistently ranks the correct molecular ion for each fragment spectrum in one of the top positions, with average ranks of 1.5, 1, and 1.2 in the three test datasets, respectively. A proof of function was obtained for a combination of CI and EI spectra, but the tool can be directly used for other soft ionization techniques such as APCI-MS.

To our knowledge, this is the first tool available to achieve such a combination of data from multiple ionization techniques, something that was previously required to be performed manually. It enables users of high-resolution GC-MS instrumentation that rely on electron ionization spectra for their analysis to add a molecular ion-generating technique to their annotation pipeline. MSdeCIpher automates and streamlines this process and thus paves the way to sophisticated compound identification tools working with GC-HRMS data. Candidate molecular ions and fragment spectra can be used directly as input for other tools capable of compound annotation in GC-HRMS, such as MS-FINDER [9]. With the option to import input data and export results in accessible comma separated files, MSdeCIpher can be integrated into existing and future data deconvolution and annotation pipelines.

## Data Availability

Metabolomics dataset from *Skeletonema costatum* can be obtained at ebi.ac.uk/metabolights/editor/MTBLS1104. The source code, a tutorial for MSdeCIpher, and an example dataset can be obtained at github.com/Pohnert-Lab/MSdeCIpher (accessed 21 December 2023).

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
