# Peer review of "MSdeCIpher: A Tool to Link Data from Complementary Ionization Techniques in High-Resolution GC-MS to Identify Molecular Ions"

_metabolites, 2023, doi:10.3390/metabo14010010_

Round 1

Reviewer 1 Report

Comments and Suggestions for Authors

The paper deals with the application of a novel software tool to identify compounds from High resolution mass spectrometry data. The problem of identifying metabolites in real samples is actual in the field and the authors developed just another interesting R based software tool among many others. They should better highlight the advantages of their tool in the field of metabolomics.

In its present form the study has strong limitations:

11)      it is applied to too few experimental data

22)      Just one instrument (Orbitrap) data are analysed.

33)      only molecular ions obtained from soft chemical ionization techniques are analysed. The effect of soft chemical ionization on spectra is hinted in the introduction but not well developed then, to the point that advantages of the approach in the metabolomics field are obscured.

44)      The application on metabolites is very limited and this is a particularly limiting aspect for a journal dedicated to metabolomics

55)      Lacks whatever comparison performance with other software tools available in the research field and on the market

66)      Pseudospectra are analyzed, without explaining what these are in fact

77)      Figure 4 shows spectra that are not well detailed on the x axis.  More focused image spectra could better explain the concept illustrated.

Comments on the Quality of English Language

understandable

Author Response

11)      it is applied to too few experimental data

See next answer

22)      Just one instrument (Orbitrap) data are analysed.

The scope of this manuscript being a Technical Note, we introduce the new concept and provide a solution that is universally applicable and can be integrated into other routines. We feel that the amount of experimental data provided is sufficient given the intention of a technical note.

33)      only molecular ions obtained from soft chemical ionization techniques are analysed. The effect of soft chemical ionization on spectra is hinted in the introduction but not well developed then, to the point that advantages of the approach in the metabolomics field are obscured.

We state that, for identification of unknowns, accurate molecular masses are required but EI spectra are often devoid of molecular ions. Therefore, the need for soft ionization techniques is established.

We have amended the manuscript to include a short explanation of soft ionization (line 47). That, together with the references cited, should provide readers unfamiliar with soft ionization with sufficient material to illustrate the concept of the manuscript.

44)      The application on metabolites is very limited and this is a particularly limiting aspect for a journal dedicated to metabolomics

We do not understand this point. We have cited a multitude of metabolomics studies in which the combination of EI and soft ionization spectra has yielded results (references 14, 17-20). Two of the three test datasets deal exclusively with metabolites, one drawn from actual cell extracts.

55)      Lacks whatever comparison performance with other software tools available in the research field and on the market

Our tool provides a unique aspect (connection of EI fragments to molecular ions from a soft ionization datasets) that, to our knowledge, no other freely available tool currently provides. We have now amended our manuscript to highlight that fact (line 16, 380). While it is true we do not provide a comprehensive comparison of common aspects of our tool with other tools out there, this hasn’t been the goal of our study. We didn’t aim for a tool that outperforms or is more comprehensive than other existing tools. Rather, we aimed to provide a very specialized solution for a group of users (those who need the unique aspect of combining EI and soft ionization data) but also to provide the new ideas contained within to the community to pave the way towards comprehensive compound characterization in GC-MS (amended at line 384). This is, in our opinion, perfectly in scope for a technical note.

66)      Pseudospectra are analyzed, without explaining what these are in fact

Thank you for your feedback. We have now noted what pseudospectra are at their first appearance in the manuscript (line 140).

77)      Figure 4 shows spectra that are not well detailed on the x axis.  More focused image spectra could better explain the concept illustrated.

We are unsure why focus on the x-axis would benefit the message of the figure. The colour coding of the fragments provides a clear message. Knowing the exact m/z for all fragments is, in our mind, unnecessary to the point the figure is trying to make.

Reviewer 2 Report

Comments and Suggestions for Authors

The manuscript submitted by D. Stettin and G. Pohnert describes development of an R-based application intended on peak search and annotation from sets of signals obtained in EI and CI HRMS. The developed software can be helpful in the analysis of metabolomics data.

The manuscript is very well written, it describes the algorithm clearly with corresponding examples. So, the developed tool can be used by specialists dealing with mass spectrometry as a tool in different area.

I have only following minor comments to correct in the manuscript:

1) Page 6, line 186: Figure 3 mentioned in the text is absent.

2) Page 7, line 219: the variable Intrel is likely to be typed with "rel" as a subscript.

3) Table 1 and Table 3: Please correct MeoX as MeOX.

Author Response

The manuscript submitted by D. Stettin and G. Pohnert describes development of an R-based application intended on peak search and annotation from sets of signals obtained in EI and CI HRMS. The developed software can be helpful in the analysis of metabolomics data.

The manuscript is very well written, it describes the algorithm clearly with corresponding examples. So, the developed tool can be used by specialists dealing with mass spectrometry as a tool in different area.

Thank you for your comments.

I have only following minor comments to correct in the manuscript:

1) Page 6, line 186: Figure 3 mentioned in the text is absent.

Figure 3 is indeed missing from the manuscript. The missing figure has been added (line 216).

2) Page 7, line 219: the variable Intrel is likely to be typed with "rel" as a subscript.

Thank you, it is now in correct subscript (line 259).

3) Table 1 and Table 3: Please correct MeoX as MeOX.

Thank you, it has been corrected.

Reviewer 3 Report

Comments and Suggestions for Authors

The manuscript titled "MSdeCIpher: A Tool for Integrating Data from Complementary Ionization Techniques in GC-MS for Molecular Ion Identification" makes a significant contribution to the mass spectrometry field by introducing MSdeCIpher, a software tool designed to integrate and analyze data from gas chromatography-mass spectrometry (GC-MS) datasets acquired using various ionization techniques. MSdeCIpher aims to automate the identification and assignment of molecular ions to fragment spectra obtained from both electron ionization (EI) and chemical ionization (CI) methods. I recommend acceptance pending the resolution of the following minor issues:

(1)    While the materials and methods section is thorough in providing details on analytical standards, sample preparation, and data acquisition, it would benefit from additional clarity regarding the specific settings and parameters used in MSdeCIpher, especially in the data deconvolution process.

(2)    The results section effectively showcases the tool's performance on metabolite standards and a real-world metabolomics dataset, offering valuable insights into MSdeCIpher's efficacy through metrics like the average rank of correct molecular ions and the successful identification rate. However, a more comprehensive discussion of the limitations and challenges encountered during the evaluation would enhance the manuscript's transparency.

Comments on the Quality of English Language

The language is technically proficient.

Author Response

The manuscript titled "MSdeCIpher: A Tool for Integrating Data from Complementary Ionization Techniques in GC-MS for Molecular Ion Identification" makes a significant contribution to the mass spectrometry field by introducing MSdeCIpher, a software tool designed to integrate and analyze data from gas chromatography-mass spectrometry (GC-MS) datasets acquired using various ionization techniques. MSdeCIpher aims to automate the identification and assignment of molecular ions to fragment spectra obtained from both electron ionization (EI) and chemical ionization (CI) methods. I recommend acceptance pending the resolution of the following minor issues:

(1)    While the materials and methods section is thorough in providing details on analytical standards, sample preparation, and data acquisition, it would benefit from additional clarity regarding the specific settings and parameters used in MSdeCIpher, especially in the data deconvolution process.

Thank you for your feedback. We have extended the section and provided an overview of deconvolution parameters in the supplementary (line 146). We have also further clarified that MSdeCIpher does not cover any data deconvolution and that it’s possible for users to adapt any of their existing deconvolution pipelines for MSdeCIpher. Nevertheless, users should be able to replicate the deconvolution pipeline that was used in the manuscript (based on XCMS and CAMERA). We are confident that we have now achieved sufficient clarity in the manuscript for that (entire section 2.4, line 135)

As far as MSdeCIpher parameters are concerned, we have included a mention of the MSdeCIpher tutorial that provides user-friendly explanations for all parameters (line 166), in addition to the more thorough examination of the MSdeCIpher workflow provided in the “Workflow” section of the manuscript.

(2)    The results section effectively showcases the tool's performance on metabolite standards and a real-world metabolomics dataset, offering valuable insights into MSdeCIpher's efficacy through metrics like the average rank of correct molecular ions and the successful identification rate. However, a more comprehensive discussion of the limitations and challenges encountered during the evaluation would enhance the manuscript's transparency.

Thank you for your feedback. We have restructured the discussion and highlighted the limitations we encountered when evaluating the tool (Chapter 4, line 352).

Reviewer 4 Report

Comments and Suggestions for Authors

The authors have prepared a Technical Note, but upon reviewing the instructions for authors, I see that the journal "Metabolites" only accepts Articles, Reviews, Protocols, and Data Descriptors. Reading the manuscript, it is more of a protocol, so it should be adjusted accordingly.

Generally, when proposing an acronym such as MSdeCIpher, it should be clarified where these letters come from. I assume MS stands for Mass Spectrometry and CI for Chemical Ionization, but the acronym needs to be clarified.

What advantages does your proposed tool have over HRMS and EI analysis? For example, the search for molecular ions in high resolution. How would this be resolved in linear chain alkanes, which tend to have the same fragmentation pattern and loss of the molecular ion? Such examples need to be included in the publication, where the power of the tool you have developed should be evident.

The abstract generally describes the process and basis of the tool. However, it overlooks why there is a need to use this tool and what its advantages are over the software provided by each equipment manufacturer. For example, the software from Agilent, Thermo, and Waters tend to be very comprehensive, where one can elucidate these issues. In this case, for what audience or research is your software intended? Please clarify these ideas in your abstract.

The tool you propose seems exciting and could be very useful. However, what happens if a user only has one ionization source in their lab? Could your tool discern between some conserved ions and the molecular ion in high resolution? Or vice versa, conserved ions that align with a molecular ion in high resolution? Clarify this with an example in your manuscript.

I think the above comments should be clarified in the introduction. Also, the objective and scope of the tool need to be clearly presented. For example, I see excellent potential for your tool in the pharmaceutical and food industry, and no mention is made of this nor biochemical markers in diseases of global interest like cancer, diabetes, etc. Therefore, I suggest the authors restructure the introduction, giving the basic principles of the tool, the coding language it was programmed in, the theorem of how it resolves uncertainties and analyzes MS spectra in low and high resolution, and finally, its applications of interest.

Take the time to improve the resolution of Figure 1, as it appears pixelated.

Scheme I has errors marked by the software used; that image was copied and pasted. Use specialized imaging software to improve the resolution and avoid such mistakes.

As this is a protocol, a discussion section may not be necessary. Again, this section has added a description of the results, and it is understood that there is no topic to compare due to the novelty of your platform. In this sense, I recommend eliminating this section and integrating it with the results, as well as better structuring a conclusion highlighting your results and the application of the tool.

The work presented by the authors is fascinating and meets the journal's scope. Some formatting issues must be corrected and adapted to a protocol rather than a Technical Note. The results section is correct. However, the discussion is unsuitable for this publication type and requires a conclusion. The introduction needs restructuring. For now, I recommend major revisions before acceptance.

Author Response

Reviewer 4:

The authors have prepared a Technical Note, but upon reviewing the instructions for authors, I see that the journal "Metabolites" only accepts Articles, Reviews, Protocols, and Data Descriptors. Reading the manuscript, it is more of a protocol, so it should be adjusted accordingly.

While it is true that Technical Notes are not mentioned in the author instructions for Metabolites, the publisher MDPI accepts Technical Notes:

https://www.mdpi.com/about/article_typesme

“Technical notes are brief articles focused on a new technique, method, or procedure. These should describe important modifications or unique applications for the described method. Technical notes can also be used for describing a new software tool or computational method. The structure should include an Abstract, Keywords, Introduction, Materials and Methods, Results, Discussion, and Conclusions, with a suggested minimum word count of 3000 words.”

A few Technical Notes have thus far appeared in Metabolites, for example:

https://doi.org/10.3390/metabo13101052
https://doi.org/10.3390/metabo13080962
https://doi.org/10.3390/metabo13010075

Generally, when proposing an acronym such as MSdeCIpher, it should be clarified where these letters come from. I assume MS stands for Mass Spectrometry and CI for Chemical Ionization, but the acronym needs to be clarified.

Thank you for the suggestion. We have included an explanation for the acronym in the introduction (line 83).

What advantages does your proposed tool have over HRMS and EI analysis? For example, the search for molecular ions in high resolution. How would this be resolved in linear chain alkanes, which tend to have the same fragmentation pattern and loss of the molecular ion? Such examples need to be included in the publication, where the power of the tool you have developed should be evident.

As stated in the manuscript, EI spectra are often devoid of molecular ions all together. If there is no match in fragment spectra databases, this puts identification of unknowns to a dead stop, even with high resolution MS – we have hit this limitation in a previous study. We have amended the manuscript to include this information. (line 45)

In the case of alkanes: As you state correctly, molecular ions are not visible in EI and fragmentation is similar, making unambiguous identification difficult. However, they also behave differently in positive methane CI than most analytes, showing low-intensity [M-H]+ molecular ions and no clear adduct/neutral loss pattern, which makes identification with MSdeCIpher difficult. We didn’t include those as examples in the manuscript because, for linear alkanes, in our opinion the best way of identification in GC is still using retention time indices.

However, there are probably multiple use cases /compound classes where the power of our tools is very evident, which we are simply not aware of. We have put a focus on metabolites in this Technical Note, because of the scope of the journal and our own experience with metabolomics research, as well as recent research in the metabolomics community pointing in the direction our tool is aiming at, for example reference [30]. We do hope that upon publication, more use cases for the tool will become evident and pursued by researchers.

The abstract generally describes the process and basis of the tool. However, it overlooks why there is a need to use this tool and what its advantages are over the software provided by each equipment manufacturer. For example, the software from Agilent, Thermo, and Waters tend to be very comprehensive, where one can elucidate these issues. In this case, for what audience or research is your software intended? Please clarify these ideas in your abstract.

Thank you for your feedback. In fact, none of the vendors tools offers a way to combine two datasets coming from different ionization techniques into a single one and cross-match molecular ions and matching fragments via sum-formula calculation. To our knowledge, every other tool currently available only lets the user handle one ionization technique at a time. We have amended that fact in the manuscript (line 16, 380).

The tool you propose seems exciting and could be very useful. However, what happens if a user only has one ionization source in their lab? Could your tool discern between some conserved ions and the molecular ion in high resolution? Or vice versa, conserved ions that align with a molecular ion in high resolution? Clarify this with an example in your manuscript.

Unfortunately, MSdeCIpher only enables user to work with two different ionization techniques and is made for user wishing to do specifically that. Most common GC-MS are equipped to provide both data sets. As you have correctly stated in your previous point, dealing with one dataset alone is something that is covered by vendor tools in a much more comprehensive way.

I think the above comments should be clarified in the introduction. Also, the objective and scope of the tool need to be clearly presented. For example, I see excellent potential for your tool in the pharmaceutical and food industry, and no mention is made of this nor biochemical markers in diseases of global interest like cancer, diabetes, etc. Therefore, I suggest the authors restructure the introduction, giving the basic principles of the tool, the coding language it was programmed in, the theorem of how it resolves uncertainties and analyzes MS spectra in low and high resolution, and finally, its applications of interest.

Thank you for the suggestions. Like we stated previously, we are unsure as of now about all possible applications beyond metabolomics research and residue analysis. Because of this, we have refrained from making definitive statements about the applicability of our tool without proof/ additional examples. However, we have amended the introduction to include possible fields of application as per your suggestion (line 32).

We have also added a note about the intended target audience (line 87), the coding language (line 81) and the basic principle of how the tool elucidates uncertainties (line 72) in the introduction. To avoid confusion about how the tool deals with low-resolution data, we have amended the title of the manuscript, to make it clear the tool is not designed for low-resolution data.

Take the time to improve the resolution of Figure 1, as it appears pixelated.

Thank you for your feedback, the resolution has been improved.

Scheme I has errors marked by the software used; that image was copied and pasted. Use specialized imaging software to improve the resolution and avoid such mistakes.

Thank you, the error has been corrected.

As this is a protocol, a discussion section may not be necessary. Again, this section has added a description of the results, and it is understood that there is no topic to compare due to the novelty of your platform. In this sense, I recommend eliminating this section and integrating it with the results, as well as better structuring a conclusion highlighting your results and the application of the tool.

Since a Technical Note allows for a discussion and we feel that a discussion is suitable to discuss the potential caveats when using MSdeCIpher, we have kept it for now but in a restructured form (line 353). We have also added a conclusion highlighting the potential applications of the tool, being integrated into existing and future compound annotation pipelines. (line 373)

The work presented by the authors is fascinating and meets the journal's scope. Some formatting issues must be corrected and adapted to a protocol rather than a Technical Note. The results section is correct. However, the discussion is unsuitable for this publication type and requires a conclusion. The introduction needs restructuring. For now, I recommend major revisions before acceptance.

Round 2

Reviewer 1 Report

Comments and Suggestions for Authors

All previous criticisms have been addressed by the authors. The manuscript is much now more understandable and framed in context.  

Comments on the Quality of English Language

english is fine and just a few minor spelling corrections are needed

Reviewer 4 Report

Comments and Suggestions for Authors

The authors have responded to my comments, and each response was appropriately addressed. Moreover, they have made the modifications indicated in my comments to their document, for which I am grateful. There are typographical errors that should be addressed in the editorial editing. For now, I recommend accepting the manuscript for publication in Metabolites.